# Personalized Indicator Thrombocytosis Shows Connection to Staging and Indicates Shorter Survival in Colorectal Cancer Patients with or without Type 2 Diabetes

**DOI:** 10.3390/cancers12030556

**Published:** 2020-02-28

**Authors:** Zoltan Herold, Magdolna Herold, Julia Lohinszky, Magdolna Dank, Aniko Somogyi

**Affiliations:** 12nd Department of Internal Medicine, Semmelweis University, Szentkiralyi u. 46., H-1088 Budapest, Hungary; herold.magdolna@med.semmelweis-univ.hu (M.H.); lohinszky.julia@med.semmelweis-univ.hu (J.L.); somogyi.aniko@med.semmelweis-univ.hu (A.S.); 2Oncology Center, Semmelweis University, Tomo u. 25-29., H-1083 Budapest, Hungary; titkarsag.dank@med.semmelweis-univ.hu

**Keywords:** colorectal cancer, platelets, thrombocytosis, survival analysis

## Abstract

*Background:* Pre- and postoperative thrombocytosis was reported to have significant effect on patient survival. However, the definition of thrombocytosis throughout the literature is not unified. *Methods:* A retrospective longitudinal observational study has been conducted with the inclusion of 150 colorectal cancer (CRC) patients and 100 control subjects. A new measure of platelet changes at an individual level, named personalized indicator thrombocytosis (PIT) was defined, including 4 anemia adjusted variants. *Results:* In concordance with the literature, PIT values of control subjects showed a slow decrease in platelet counts, while PIT values of CRC patients were significantly higher (*p* < 0.0001). More advanced staging (*p* < 0.0001) and both local (*p* ≤ 0.0094) and distant (*p* ≤ 0.0440) metastasis are associated with higher PIT values. Higher PIT values suggested shorter survival times (*p* < 0.0001). Compared to conventional, a PIT-based definition resulted in approximately 3-times more patients with thrombocytosis. 28% and 77% of the deceased patients had conventional- and PIT-based thrombocytosis, respectively. *Conclusions:* Compared to conventional thrombocytosis, as an individual metric, PIT values may indicate the condition of patients more precisely. Possible future applications of PIT may include its usage in therapy decision and early cancer detection; therefore, further investigations are recommended.

## 1. Introduction

Platelets play a significant role in tumor growth and metastasis [1]. It is known since the late 1800s that gastrointestinal tumors and platelets have a strong connection [1,2]. Genetic and animal model studies revealed that platelets help tumor cells during their migration in the blood stream and protect them from the immune system caused by changes to surface receptors on both tumor- and platelet cells [1,3,4,5,6,7]. Human observational studies concluded that in many tumor types [8,9,10,11,12], patients with platelet counts over the upper normal range (thrombocytosis) have shorter survival, compared to those within the normal range. However, it must be stated that in many other tumor sites thrombocytosis could not be observed at all [13].

The cause behind thrombocytosis can be due to a number of factors including bleeding of the tumor or due to metabolic changes caused by tumor cells. The second process has been named as paraneoplastic thrombocytosis [14,15]. Thrombocytosis is known to be characteristic for gastrointestinal, including colorectal tumors [16]. Both pre- and postoperative thrombocytosis proved to be significant factors affecting survival in colorectal cancer (CRC) [16,17,18,19]. However, in the available literature, several different platelet count cut-off values have been proposed for the diagnosis of thrombocytosis associated with tumorous diseases. The most commonly used cut-off value is the upper normal limit (400 × 10^9^/L) of platelets—used by most of the laboratories, but several other values has been suggested as well, some of them were even within normal range [16].

In healthy individuals, the number of platelets decreases with age [20,21]. It has been suggested that in many applications, using a personalized platelet count might be more appropriate [22,23]. These raised our question of whether it is preferable to define some kind of personalized thrombocytosis in colorectal cancer—maybe tumors in general—and to examine its effect on survival and its connection to other oncological factors, such as staging.

To test our theory, a retrospective longitudinal observational study has been conducted to evaluate the usefulness of the newly introduced measure named personalized indicator thrombocytosis (PIT)—a ratio of platelet counts between two time points. Data were recorded at two time points: at the time of tumor diagnosis and at least 30 months prior to the onset of CRC. Further aims of our study were to search for relationships between PIT and various clinical outcomes such as TNM staging and known comorbidities. Diabetes and antiplatelet therapy were special interests; and anemia adjusted variants of PIT have been introduced and investigated as well.

Diabetes mellitus, a complex metabolic and endocrine disease, is one of the most prevalent diseases in our time, occurring in approximately 8–9% of the world’s population. Approximately 90% of diabetic patients are suffering from type 2 diabetes (T2DM). An estimated 20% of the world’s population over the age of 60 may be affected [24,25,26]. Compared to the healthy population, in T2DM an increased occurrence of malignancies is confirmed. Among T2DM patients pancreas, CRC, bladder, liver, endometrium and breast cancer is the most common [27,28,29,30]. An increased incidence of CRC is known within T2DM patients [31], which is associated with an increased risk of shorter survivals [32]. T2DM is also associated with a dysfunction of platelets [33,34], therefore, a large proportion of T2DM patients receive preventive antiplatelet therapy [35,36,37].

## 2. Results

### 2.1. Measurements at the Time of Colorectal Cancer Diagnosis

A total of 100 control subjects and 150 CRC patients were included in our study. T2DM was observed in 32% of patients with colorectal cancer. Study participants were divided into four cohorts based on the presence of T2DM: control subjects without T2DM were assigned to cohort 1, controls with T2DM to cohort 2, CRC patients without T2DM to cohort 3 and CRC patients with T2DM to cohort 4. The number of participants in each cohort was 50, 50, 102 and 48, respectively.

Subjects in cohort 1 were slightly younger, compared to all other cohorts (*p* ≤ 0.0089). Between subjects of cohort 1 and 2 no differences were observed in complete blood count parameters, however, hypertension (*p* = 0.0001) and the usage of antiplatelet agents (*p* < 0.0001) was higher in diabetic control subjects (Table 1).

Comparison between cohorts 3 and 4 revealed that both hemoglobin (*p* = 0.0330) and hematocrit (*p* = 0.0348) values of CRC patients with T2DM are significantly higher (Table 1). Almost every patient within cohort 4 had hypertension, compared to the 74.5% in cohort 3 (*p* = 0.0030). CRC patients without T2DM had significantly less major cardiovascular events before the onset of the tumor (*p* = 0.0315), and usage of antiplatelet therapy was less common, compared to CRC patients with T2DM (*p* = 0.0030). No difference was found in mean survival times (*p* = 0.4002) between the two cancer cohorts (Table 1).

Comparison between tumor and control subjects was performed in all possible combinations. As expected, complete blood count results of CRC patients were shifted, while control patients’ results were normal. Comparisons of anamnestic data reflected the differences in age and know comorbidities of the study subjects (Table 1).

### 2.2. Pre-Tumor Complete Blood Count Measurements

Pre-tumor complete blood count measurements was preformed 9.98 ± 2.43 years (cohort 1), 7.46 ± 2.74 years (cohort 2), 6.80 ± 2.74 years (cohort 3) and 7.34 ± 5.94 years (cohort 4) prior to CRC diagnosis. Comparisons between cohorts were performed in all possible combinations: no differences were found (Table 2).

### 2.3. Personalized Indicator Thrombocytosis

Personalized indicator thrombocytosis (PIT), and its hemoglobin- (PIT_Hgb_), hematocrit- (PIT_Htc_), red blood cell- (PIT_RBC_) and mean corpuscular volume (PIT_MCV_) adjusted versions were calculated for every study participant, as described in Methods (Equations (1)–(5)). Individuals with CRC had higher values of every PIT variants, compared to controls (*p* < 0.0001). Diabetes had no impact on PIT variants neither in tumor (*p* ≥ 0.1762), nor in control (*p* = 0.9999) cohorts (Figure 1, Appendix A).

To obtain the optimal cut-off points of the various PIT variants—that may very likely indicate the presence of colorectal cancer, receiver operating characteristic (ROC) analyses have been performed. A significant proportion of the study participants received antiplatelet treatment (40.8%), therefore, at first the ROC curves of treated and non-treated groups have been compared. DeLong’s test for two ROC curves indicated no difference between the treated and not treated groups in any of the PIT variants (*p* ≥ 0.3146), therefore, all subsequent ROC analyzes reported below were performed with all 250 study subjects.

To detect colorectal cancer, PIT had 93.0% specificity and 66.7% sensitivity with an optimal cut-off point of 1.12 (Figure 2a). Adjusted variants of PIT had specificity between 90.0% and 97.0%, sensitivity between 63.3% and 74.0%, and optimal cut-off points were estimated between 1.12 and 1.20 (Figure 2b–e). Defining thrombocytosis with the resulting lowest and highest cut-off points, compared to the “conventional” diagnosis (platelet count > 400 × 10^9^/L), thrombocytosis can be observed in at least 100 (66.7%, cut-off point: 1.12, *p* < 0.0001) and 83 (55.3%, cut-off point 1.20, *p* < 0.0001) patients, respectively. If thrombocytosis is defined with the mean + 2 standard deviation PIT value of healthy control subject, thrombocytosis is present in 87 patients (58%, cut-off point: 1.17, *p* < 0.0001). Conventional thrombocytosis can be observed in 34 (22.7%) CRC patients (Appendix A).

Random intercept linear mixed effect models were constructed to determine the annual changes of platelets, platelet/hemoglobin-, platelet/hematocrit-, platelet/red blood cell (RBC) count- and platelet/mean corpuscular volume (MCV) ratios. Dependent variables were the natural logarithmic value of platelet and its anemia adjusted variants listed above, fixed effect was the time between the two laboratory measurements and patient IDs were used as random effect. Whereas in control subjects a slight but continuous decrease was seen, in CRC patients a steady rise of the parameters were observed (Table 3).

#### 2.3.1. Relationship of Personalized Indicator Thrombocytosis to other Grouping Variables

It was tested whether other factors such as sex, chemotherapy, antiplatelet therapy, known comorbidities and cancer staging affects any of the PIT values. All of the PIT variants were the same in both sexes (*p* ≥ 0.3660) and no differences was found neither if patient received chemotherapy or not (*p* ≥ 0.2050), nor if patient received antiplatelet therapy or not (*p* ≥ 0.0885). None of the comorbidities had detectable effect on PIT values.

With the increasing size of the primary tumor all PIT variants are increasing as well (*p* < 0.0001). Multiple comparisons suggest that stages T1 and T2 are very similar (*p* ≥ 0.4750), but in the case of T2 range is much wider. PIT values in stages T3 and T4 do not differ (*p* ≥ 0.1283) as well (Appendix A, Figure 3).

Compared to the conventional definition, with the usage of PIT > 1.12, the occurrence of thrombocytosis is the same in stage T1 (conventional vs. PIT > 1.12: 3 vs. 3 cases, *p* = 1.0000), but in stage T2 (2 vs. 10 cases, *p* = 0.0469) and T3 (13 vs. 60 cases, *p* < 0.0001) the occurrence is significantly higher in the case of PIT > 1.12. In stage T4 the occurrence is higher with PIT > 1.12 but just marginally (6 vs. 13 cases, *p* = 0.0513); and in the 14 unresectable cases higher, but statistically not justifiable occurrences were observed (8 vs. 13 cases, *p* = 0.1011).

Significantly higher number of CRC related deaths were observed with PIT > 1.12 (16 vs. 44 cases, *p* < 0.0001). No difference can be observed between patients with (death without thrombocytosis: 3 cases; death with thrombocytosis: 15) and without antiplatelet therapy (death without thrombocytosis: 10 cases; death with thrombocytosis: 29 cases; *p* = 0.6810).

Degree of spread to regional lymph nodes and the presence of distant metastasis has been also analyzed (Appendix A, Figure 4). Patients without regional lymph node metastasis have significantly lower PIT values than patient who have at least N1 lymph node metastasis (*p* ≤ 0.0094). PIT (*p* = 0.0140), PIT_RBC_ (*p* = 0.0360) and PIT_MCV_ (*p* = 0.0440) have significantly higher values if distant metastasis is present.

#### 2.3.2. Survival Analysis of Personalized Indicator Thrombocytosis

Two endpoint events have been defined: 1. death related to colorectal cancer and 2. lost to follow-up (LFU). Death related to colorectal cancer occurred in 57 (38.0%), while LFU occurred in 8 (5.3%) cases. The effect of PIT variants and diabetes on survival was investigated. Due to the two endpoints, competing risk models were used. For the sake of clarity, the results of the survival models are presented as follows: since the LFU event was used only as a “correction factor” for the more accurate determination of colorectal cancer mortality; therefore, the results for the event “death related to colorectal cancer” are reported only.

Diabetes had no effect on survival in any models (*p* ≥ 0.2103). PIT had a hazard ratio (HR) of 3.14 (95% confidence interval: 2.05–4.81; *p* < 0.0001). A HR of 1.64, 1.96, 2.71 and 2.03 was observed for PIT_Hgb_, PIT_Htc_, PIT_RBC_ and PIT_MCV_ (*p* < 0.0001), respectively (Figure 5).

In the everyday clinical practice, a rise in platelet counts of less than 100% is expected with a much higher probability. Therefore, hazard ratios for 10% elevations of platelet changes has been calculated as well, which was achieved by a technical step (multiplication of PIT values by 10). A HR of 1.12, 1.05, 1.07, 1.10 and 1.07 was observed for every 10% increase of PIT, PIT_Hgb_, PIT_Htc_, PIT_RBC_ and PIT_MCV_ (*p* < 0.0001), respectively (Appendix A).

## 3. Discussion

In healthy individuals, platelet counts range between 280 ± 130 × 10^9^/L [38]. While in infants and children this number may be higher; and over 60 years of age it is usually lower [20,21,39,40,41]. With aging not only numerical but functional changes also occur to platelets [20,21], for example platelet activity is increased. These changes may be due to increased oxidative stress, age-related mRNA and microRNA changes and various comorbidities. It has been shown that in many diseases, such as solid tumors and diabetes mellitus, both quantitative and qualitative changes of platelets may occur [1,3,4,5,6,7,8,9,10,11,12,42,43]. In accordance with the literature, control subjects in our study had decreasing platelet count with age, while in CRC patients the opposite showed: a significant increase of platelets can be found between pre-tumor and at-the-diagnosis blood sampling results.

Thrombocytosis, the elevation of platelet count is common in various diseases [15]. It has been observed in malignancies with various locations as well [8,9,10,11,12], including gastrointestinal cancers [16,44]. However, it is important to note that thrombocytosis has been defined with several thresholds in those publications [16]. The mechanism behind the platelet changes in malignancies is called paraneoplastic thrombocytosis [45]. Based on the results available in the literature, the proportion of gastrointestinal cancer patients with thrombocytosis varies from a few percent to approximately 50% [16]. Thrombocytosis was reported to have a prognostic role both prior to and after primary tumor removal [16,17,18,19]. Furthermore, studies on CRC and other platelet-related indicators and ratios like mean platelet volume (MPV), the mean platelet volume—platelet count ratio (MPV/PC) and the platelet - lymphocyte ratio (PLR) were conducted recently [19,46,47]. Compared to the healthy controls, CRC patients had significantly lower MPV/PC and significantly higher PLR values [46]. Worse PLR and mean platelet values were reported in more advanced stages [48]. Decreasing MPV values following a whole course of treatment predicts poorer survival [47].

In this study, we have defined a new platelet-based ratio, PIT, which provides individualized information about how much platelets have changed, relative to the time before the onset of the tumor. Adjusted variants of PIT on the various forms of anemia were defined as well. Compared to MPV/PC and PLR, all PIT variants had a slightly worse sensitivity (between 63.3–74%), but a much better specificity (between 90–97%) when distinguishing between CRC patients and controls. MPV/PC and PLR were reported to have sensitivity over 80% and specificity around 65% [46]. Association between tumor progression and increasing PIT values were found, similarly to that observed for PLR [48]. Pre- and postoperative thrombocytosis was shown to have a negative effect on patient survival [16,17,18,19,49]. PIT values had the same effect, patient survival was worse if PIT values were higher.

It was suggested previously that over 60 years of age, personalized ranges of platelets may be more accurate to avoid a misdiagnosis of thrombocytopenia [23]. We have demonstrated that in the case of thrombocytosis-associated CRC the usage of the same individualized practice may be more advisable. With the current, “conventional” thrombocytosis definition (> 400 × 10^9^/L [16]) approximately every fifth of our patients had thrombocytosis, while with using the optimal cut-off values from our ROC analysis and defining thrombocytosis with PIT values greater than 1.12, 3-times more patient had thrombocytosis-associated CRC. Comparing the two definitions with regard to staging and deaths related to CRC, we believe that the usage of a PIT-based thrombocytosis definition may be more accurate. Our theory is that higher PIT changes may be a better marker of worse outcome: Assuming a cancer milieu that cause only smaller increase in platelet count, the course of the disease is expected to be more favorable, whereas in a case where the change is more significant, meaning the tumor is more aggressive, the disease may progress more rapidly. On possible justification of this hypothesis may be that from the 57 patients, who died during the time of our investigation, 44 of them (77.2%) had PIT- based thrombocytosis, while using the conventional thrombocytosis definition the same proportion was significantly lower (16 patients, 28.1%). For the part of our hypothesis that larger changes may predict worse disease progression, the present study does not provide sufficient information, therefore, in the future, it is of great importance to carry out a study with a much larger sample size.

Based on our results and the hypothesis above, there may be several possible future applications of PIT values. One of these could be for example to examine the role of PIT values in CRC treatment decision: Current protocols recommend chemotherapy for stage III, IV and recurrent CRCs, but for stage II adjuvant chemotherapy still remains controversial [50,51,52]. Currently, there is no clear evidence if there is a benefit of adjuvant chemotherapy in stage II CRC [53]. Results from clinical trials [54,55] suggests that compared to surgery alone, adjuvant chemotherapy improved the cure and decreased the risk of death. Recurrence rate for stage II is lower, compared to higher stages [56], but to date, no good marker has been found that may predict recurrences [57,58,59]. Although the present study could not provide data to supporting this hypothesis, but in our opinion, at the diagnosis of CRC or after primary surgery, when all staging data are available, calculating the PIT value could provide some guidance for future therapeutic decisions. Whereas the location of oncological- and primary care may vary significantly, a collaboration between general practitioner and oncologist may be essential in defining PIT.

Another hypothesized future application could be the role of PIT in the early detection of CRC. Current early detection methods [60] include both non-invasive methods like (immunochemical) fecal occult blood test, radiological examination methods, fecal DNA testing and a PCR-based serum blood testing of methylated septin-9. Invasive tests include sigmoidoscopy and total colonoscopy. Most guidelines and screening programs recommend annual testing of non-invasive techniques, and invasive methods less frequently (for example every 5 years), however, the techniques used and their frequency vary from country to country [60,61,62]. In spite of the various screening programs, a significant part of the population still does not attend [63,64]. (Immunochemical) fecal occult blood test has been criticized previously because bleeding usually occurs in later development stage [65], while other techniques like genetic and PCR screening methods would present a significant financial burden on health care systems. Primary care plays the most important role in early CRC detection [66,67,68,69], and PIT values may expand their repository of diagnostic tools either by supplementing or maybe preceding current ones allowing an even earlier indication, which may suggests primary care professionals to screen for CRC.

Type 2 diabetes had a prominent role in our research as CRC develops more often in T2DM, compared to the healthy ones [30,31] and due to its known effects on platelets [42]. Almost every third CRC patients enrolled in our study also had T2DM and PIT values had slightly wider variance in both diabetic cohorts but none of them differed from the corresponding non-diabetic cohorts. PIT had the same tendency in non-cancer diabetic participants like it was observed in the control cohort, platelets are decreasing with age. Several shared molecular, genetic and environmental risk factors are known in the pathomechanism of T2DM and CRC, for example obesity, decreased vitamin D- and increased insulin like growth factor 1 serum concentration, older age, increase of inflammatory pathways, epigenetic changes, etc. [27,28,31]. It was previously demonstrated that T2DM has a negative impact on the survival of CRC patients [31,32,70]. In addition to the differences known in the literature, we could not confirm any new, T2DM specific differences.

In summary, PIT may be an important factor in the detection, treatment and management of CRC patients. Our results showed that more advanced tumor stages might be well characterized by higher PIT values. Due to its possible future application, we consider it important to carry out further studies on PIT and testing its use in early CRC detection and treatment decision should be investigated thoroughly. It is also recommended that the presence of PIT should be investigated in other tumor types, which are typically not associated with thrombocytosis according to our current knowledge, such as breast cancer [12,71]. Testing PIT with further parameters, such as ethnicity is advisable; and due to the smaller sample size of the present study, some of our results like the optimal cut-off values of PIT should be examined with a larger sample size as well.

## 4. Materials and Methods

This study was conducted in concordance with the WMA Declaration of Helsinki. Ethical permission was received from the Regional and Institutional Committee of Science and Research Ethics, Semmelweis University (SE TUKEB 21-13/1994, approval date of latest modification: 15^th^ January 2019) and from the Committee of Science and Research Ethics, Hungarian Medical Research Council (ETT TUKEB 8951-3/2015/EKU). Handling of patient data was in accordance with the General Data Protection Regulation issued by the European Union.

### 4.1. Patients and Study Design

A retrospective longitudinal observational study was conducted with the data available from the medical databases of the 2nd Department of Internal Medicine, Semmelweis University, Budapest and the Oncology Center of Semmelweis University, Budapest. A total of 1623 colorectal cancer patients’ data were screened. All patient attended at outpatient clinics, between 2014–2019. Exclusion criteria included any previous malignancies, known hematologic- and/or inflammatory bowel- and/or systemic autoimmune- and/or inadequately controlled thyroid diseases, usage of systemic corticosteroids 90 days prior visit date and/or erythropoiesis-stimulating agents and/or recent blood transfusion, and patients with an ECOG grade > 1. Inclusion criteria to the study required laboratory results performed at Semmelweis University at the time of CRC diagnosis. After verification of inclusion and exclusion criteria, 731 of the 1623 patient remained in the initial study population.

We had investigated what percentage of the 731 patients had non-tumor related visit(s) previously at any departments of Semmelweis University. Pre-tumor laboratory results have been found for 150 of the 731 patients that were performed at least 30 months prior to the onset of the tumor. Colorectal tumors are expected to develop within several years [72,73], however, since the rate and time of development can be influenced by various factors (such as genetic, environmental, lifestyle etc.), no exact estimation has been provided previously. Pre-tumor laboratory results were selected from outpatient visits only, which were related to chronic conditions that did not affect quality of life, but required continuous medical observation like diabetes, hypertension, asthma, etc. Hospitalizations, visits 6 months within the date of hospitalization and any visits related to cancer screening were excluded.

Data has been also investigated if pre-tumor laboratory results had to be more than 60 months before the onset of the tumor. 102 CRC patient had pre-tumor results older than 60 months. As results were basically the same to those with the 30-month limit, due to the larger sample size and the expectedly higher statistical power, in the final statistical analysis the 30-month limit was used.

48 of the 150 CRC patients had type 2 diabetes, which existed before the onset of CRC. For comparison, a selective population of 50 subject without, and 50 with type 2 diabetes were included into our study as reference population, who attended at the Metabolic Clinic of the 2nd Department of Internal Medicine, Semmelweis University, Budapest. Control subjects had to meet the same inclusion and exclusion criteria as CRC patients.

54 control subjects and 48 individuals with CRC used antiplatelet medications. To investigate if antiplatelet therapy affected study results, the complete statistical analysis was performed without these study subjects as well. The same results were obtained, therefore, patients receiving antiplatelet therapy were not excluded.

### 4.2. Clinical and Laboratory Data Measurements

Anamnestic data on hypertension, previous major cardiovascular events, thyroid diseases, cholelithiasis, appendicitis, diabetes and the usage of antiplatelet medications were collected. Complete blood count was measured at the Central Laboratory of Semmelweis University. In the case of CRC patients, staging was given by histopathological examination of surgical specimens, TNM Classification of Malignant Tumors was used. Chemotherapy was recorded as a dichotomized factor as well. Survival time was calculated from the time of CRC diagnosis until patient’s death, or until last appearance at any of the Clinics of Semmelweis University (lost to follow-up patients), or in the case of surviving patients until 31. October 2019.

### 4.3. Personalized Indicator Thrombocytosis

Personalized indicator thrombocytosis (PIT) was specified to be a useful tool to investigate the direction and magnitude of platelet changes between two given times. Computation of PIT is described in Equation (1). Hemoglobin-, hematocrit-, red blood cell count- and mean corpuscular volume adjusted versions of PIT were also determined in order to adjust to the various forms of anemia [74] (Equations (2)–(5)).

Personalized indicator thrombocytosis:(1)$$\mathrm{PIT}=\frac{{\mathrm{Platelet}\;\mathrm{count}}_{\mathrm{Time}\;2}}{{\mathrm{Platelet}\;\mathrm{count}}_{\mathrm{Time}\;1}}$$

Hemoglobin adjusted personalized indicator thrombocytosis:(2)$${\mathrm{PIT}}_{\mathrm{Hgb}}=\frac{{\left(\frac{\mathrm{Platelet}\;\mathrm{count}}{\mathrm{Hemoglobin}}\right)}_{\mathrm{Time}\;2}}{{\left(\frac{\mathrm{Platelet}\;\mathrm{count}}{\mathrm{Hemoglobin}}\right)}_{\mathrm{Time}\;1}}$$

Hematocrit adjusted personalized indicator thrombocytosis:(3)$${\mathrm{PIT}}_{\mathrm{Htc}}=\frac{{\left(\frac{\mathrm{Platelet}\;\mathrm{count}}{\mathrm{Hematocrit}}\right)}_{\mathrm{Time}\;2}}{{\left(\frac{\mathrm{Platelet}\;\mathrm{count}}{\mathrm{Hematocrit}}\right)}_{\mathrm{Time}\;1}}$$

Red blood cell (RBC) count adjusted personalized indicator thrombocytosis:(4)$${\mathrm{PIT}}_{\mathrm{RBC}}=\frac{{\left(\frac{\mathrm{Platelet}\;\mathrm{count}}{\mathrm{RBC}\;\mathrm{count}}\right)}_{\mathrm{Time}\;2}}{{\left(\frac{\mathrm{Platelet}\;\mathrm{count}}{\mathrm{RBC}\;\mathrm{count}}\right)}_{\mathrm{Time}\;1}}$$

Mean corpuscular volume (MCV) adjusted personalized indicator thrombocytosis:(5)$${\mathrm{PIT}}_{\mathrm{MCV}}=\frac{{\left(\frac{\mathrm{Platelet}\;\mathrm{count}}{\mathrm{MCV}}\right)}_{\mathrm{Time}\;2}}{{\left(\frac{\mathrm{Platelet}\;\mathrm{count}}{\mathrm{MCV}}\right)}_{\mathrm{Time}\;1}}$$

### 4.4. Statistical Analysis

Statistical analyses were performed within the R for Windows version 3.6.1 environment [75]. One-way ANOVA models with Tukey’s all-pair post-hoc tests [76] were used for comparisons between groups. For nonparametric multiple comparisons Kruskal-Wallis tests with p-value adjusted pairwise Mann-Whitney U-tests were used. Chi-squared test was used to compare count data. Receiver operating characteristic (ROC) analysis was performed to determine optimal cut-off points and to test the sensitivity and specificity of various forms of PIT. Comparison of ROC curves was performed with DeLong’s test [77]. A random intercept linear mixed effect model was used to determine the annual changes in platelets (R package nlme [78]). Cause-specific competing risk survival models were performed with the R packages survival [79], mstate [80] and forestplot [81]. *p* < 0.05 was considered as statistically significant; p-values were corrected with the false discovery rate method for multiple comparisons problem. Continuous data were reported as mean ± standard deviation, while the number of occurrences and their percentage in parentheses characterized frequency data.

Boxplots were drew with the default settings of R: the bottom and top of the box are the lower and upper quartiles; the band near the middle is the median. The upper whisker is defined as the smaller value from the maximum or upper quartile + 1.5 interquartile range, whereas the lower whisker is the larger value from the minimum or lower quartile − 1.5 interquartile range.

## 5. Conclusions

In this study, we introduced a new measure of platelet change at an individual level, named as personalized indicator thrombocytosis, PIT. To adjust the effects of various anemia forms, hemoglobin, hematocrit, red blood cell count and mean corpuscular volume adjusted PIT variants have been defined as well. PIT values indicated a decrease of platelet count in control patients with age, while in colorectal cancer patients the opposite, a significant increase could be observed. More advanced tumor stages were associated with higher PIT values, and patients with higher PIT values had shorter expected survival.

As PIT is a personalized metric, there may be many possible future applications of PIT values including its hypothesized usage in therapy decision and early cancer detection. Its close relationship with tumor progression and patient survival may support this idea, therefore, further thorough investigations are strongly recommended.

## Figures and Tables

**Figure 1 cancers-12-00556-f001:**
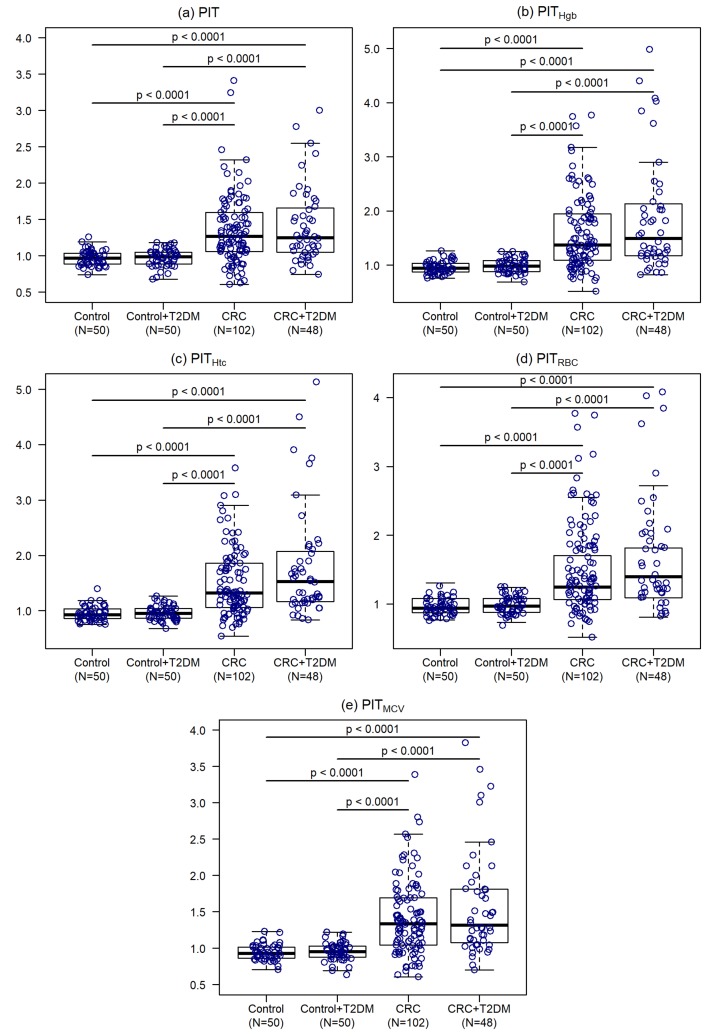
Personalized indicator thrombocytosis (**a**), hemoglobin adjusted personalized indicator thrombocytosis (**b**), hematocrit adjusted personalized indicator thrombocytosis (**c**), red blood cell adjusted personalized indicator thrombocytosis (**d**) and mean corpuscular volume adjusted personalized indicator thrombocytosis (**e**) in the 4 cohorts. CRC: colorectal cancer; T2DM: type 2 diabetes mellitus.

**Figure 2 cancers-12-00556-f002:**
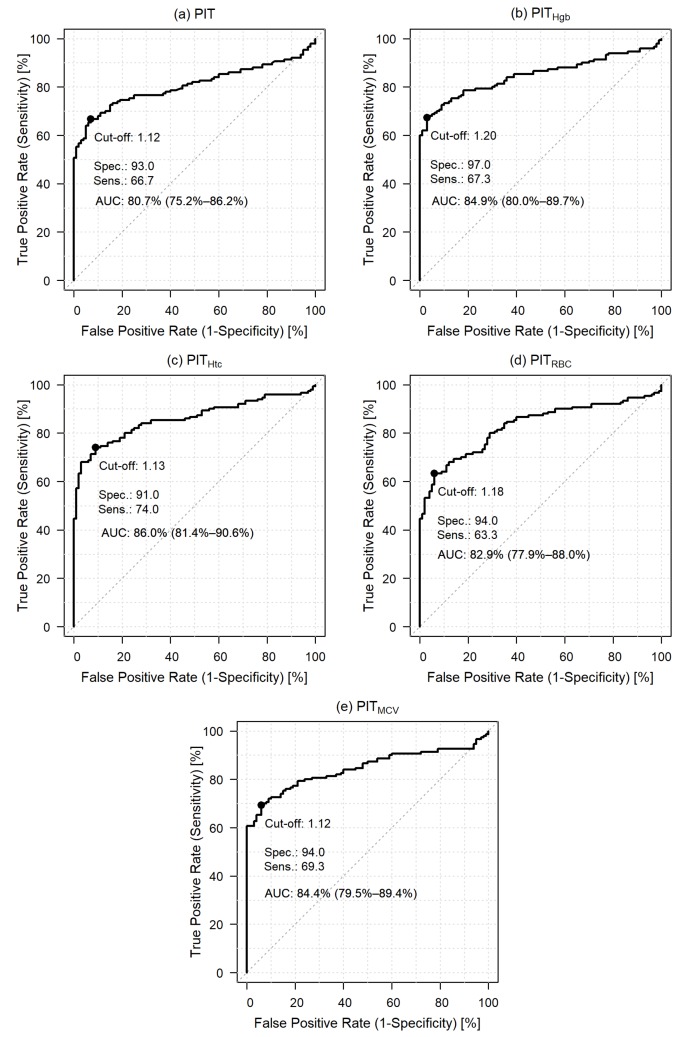
Receiver operating characteristic (ROC) curve to detect colorectal cancer based on personalized indicator thrombocytosis (**a**), hemoglobin adjusted personalized indicator thrombocytosis (**b**), hematocrit adjusted personalized indicator thrombocytosis (**c**), red blood cell adjusted personalized indicator thrombocytosis (**d**) and mean corpuscular volume adjusted personalized indicator thrombocytosis (**e**).

**Figure 3 cancers-12-00556-f003:**
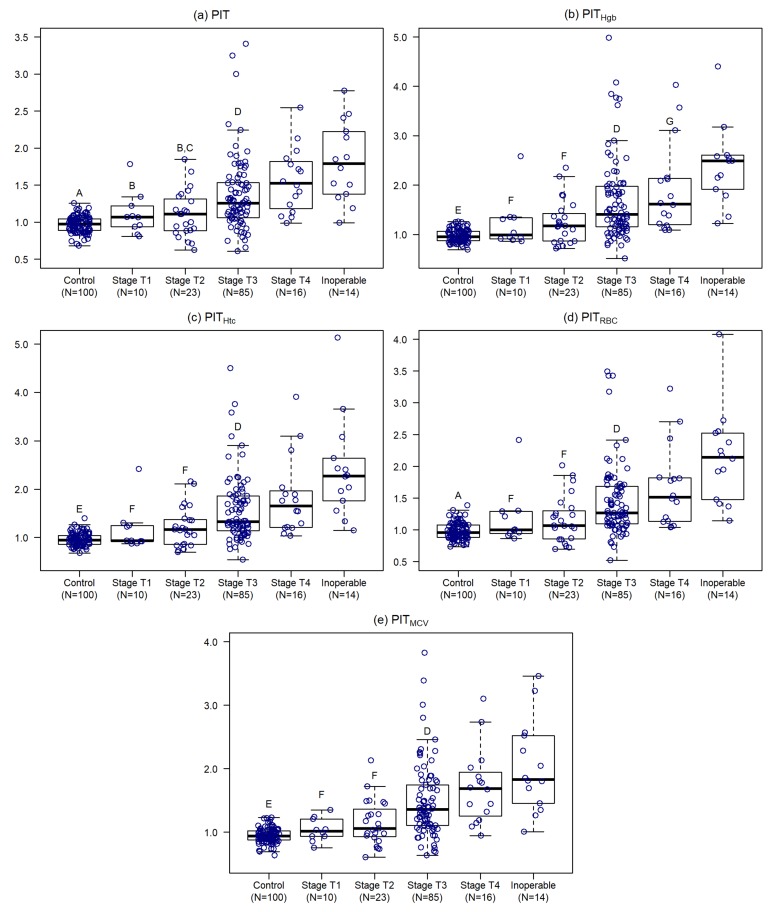
Personalized indicator thrombocytosis (**a**), hemoglobin adjusted personalized indicator thrombocytosis (**b**), hematocrit adjusted personalized indicator thrombocytosis (**c**), red blood cell adjusted personalized indicator thrombocytosis (**d**) and mean corpuscular volume adjusted personalized indicator thrombocytosis (**e**) within the different tumor sizes. ^A^ Significant difference from stage T3, T4 and inoperable (*p* < 0.001); ^B^ Significant difference from stage T4 and inoperable (*p* < 0.01); ^C^ Significant difference from stage T3 (*p* < 0.05); ^D^ Significant difference from inoperable (*p* < 0.01); ^E^ Significant difference from stage T2, T3, T4 and inoperable (*p* < 0.01); ^F^ Significant difference from stage T3, T4 and inoperable (*p* < 0.05); ^G^ Significant difference from inoperable (*p* < 0.05).

**Figure 4 cancers-12-00556-f004:**
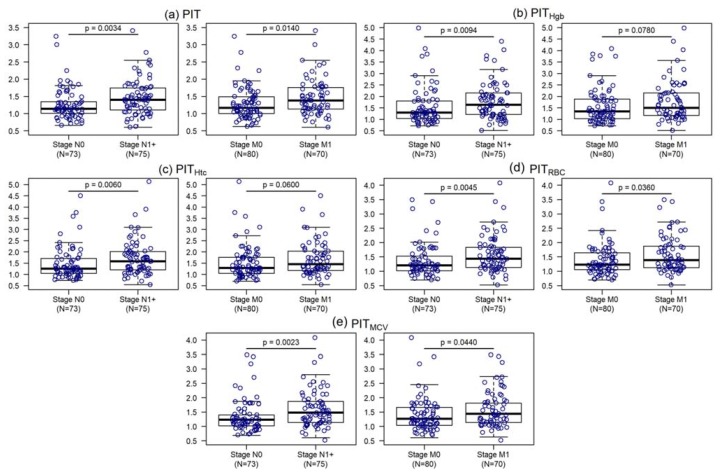
Personalized indicator thrombocytosis (**a**), hemoglobin adjusted personalized indicator thrombocytosis (**b**), hematocrit adjusted personalized indicator thrombocytosis (**c**), red blood cell adjusted personalized indicator thrombocytosis (**d**) and mean corpuscular volume adjusted personalized indicator thrombocytosis (**e**) within the different stages of degree of spread to regional lymph nodes and the presence of distant metastasis.

**Figure 5 cancers-12-00556-f005:**
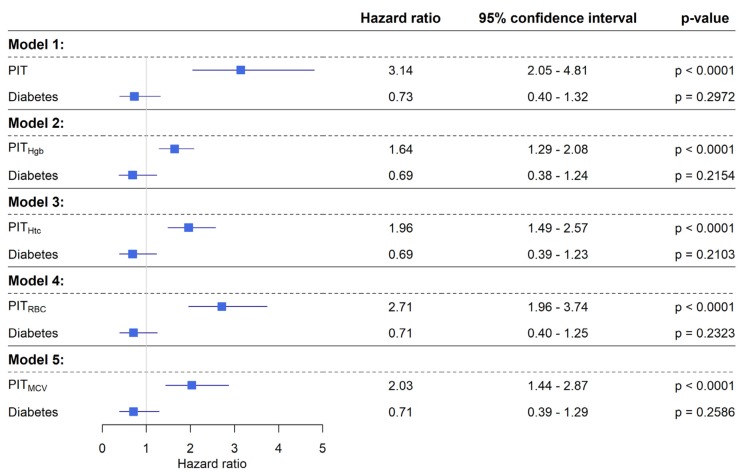
Results of five individual competing risk survival models to test the effect of PIT variants and diabetes. Diabetes had no effect on patient survival in any of the models. Every 100% increase of personalized indicator thrombocytosis (PIT) results with an increased risk of shorter survival time (*p* < 0.0001). Similarly, increased hemoglobin adjusted personalized indicator thrombocytosis (PIT_Hgb_), hematocrit adjusted personalized indicator thrombocytosis (PIT_Htc_), red blood cell adjusted personalized indicator thrombocytosis (PIT_RBC_) and mean corpuscular volume adjusted personalized indicator thrombocytosis (PIT_MCV_) values are associated with an increased hazard of shorter survival times (*p* < 0.0001).

**Table 1 cancers-12-00556-t001:** Anamnestic and complete blood count measurement data of the 4 cohorts at the time of cancer diagnosis. CRC: colorectal cancer, T2DM: type 2 diabetes mellitus. *Continuous data: mean ± standard deviation; frequency data: number of observation (percentage).*

Variables	Cohort 1 (Control) (*N* = 50)	Cohort 2 (T2DM) (*N* = 50)	Cohort 3 (CRC) (*N* = 102)	Cohort 4 (CRC + T2DM) (*N* = 48)
Age [years]	60.84 ± 12.29 ^A^	67.52 ± 9.48	68.13 ± 9.89	70.28 ± 7.33
Duration of T2DM [years]	NA	16.78 ± 8.63	NA	11.88 ± 7.52 ^1,B^
White blood cell count [10^9^/L]	6.97 ± 1.91	7.73 ± 2.03	8.22 ± 3.21	9.19 ± 5.45 ^C^
Red blood cell count [10^12^/L]	4.80 ± 0.59	4.76 ± 1.59	4.50 ± 0.57 ^D^	4.28 ± 0.63 ^E^
Platelet count [10^9^/L]	261.71 ± 64.11	248.78 ± 61.97	334.33 ± 132.21 ^E^	343.17 ± 124.39 ^E^
Hemoglobin [g/L]	144.20 ± 13.83	138.88 ± 10.76	123.68 ± 23.11 ^E^	113.40 ± 25.44 ^E,F^
Hematocrit [L/L]	0.43 ± 0.04	0.42 ± 0.03	0.38 ± 0.06 ^E^	0.35 ± 0.06 ^E,F^
Mean corpuscular volume [fL]	89.67 ± 5.18	88.43 ± 3.74	84.23 ± 7.09 ^E^	82.25 ± 8.08 ^E^
Mean corpuscular hemoglobin [pg]	30.18 ± 2.05	29.35 ± 1.58	27.43 ± 3.39 ^E^	26.43 ± 4.18 ^E^
Mean corpuscular hemoglobin concentration [g/L]	336.42 ± 9.24	325.78 ± 45.06	324.90 ± 19.54	319.75 ± 25.32 ^C^
Red blood cell distribution width [%]	13.24 ± 0.86	13.31 ± 0.90 ^2^	14.82 ± 2.92 ^3,E^	15.60 ± 2.79 ^2,E^
Mean survival time [months]	NA	NA	31.15 ± 22.53	34.63 ± 25.60
CRC related death	NA	NA	41 (40.2%)	16 (33.3%)
Sex (Female/Male)	25/25 (50.0%/50.0%)	25/25 (50.0%/50.0%)	49/53 (48.0%/52.0%)	16/32 (33.3%/66.7%)
Hypertension	26 (52.0%)	45 (90.0%) ^G^	76 (74.1%) ^H^	46 (90.0%) ^F,G^
Previous major cardiovascular event	7 (14.0%)	14 (28.0%)	22 (21.6%)	21 (43.8%) ^C,F^
Thyroid diseases	4 (8.0%)	9 (18.0%)	16 (15.7%)	8 (16.7%)
Previous cholelithiasis	3 (6.0%) ^A^	16 (32.0%)	29 (28.4%)	19 (39.6%)
Previous appendicitis	5 (10.0%)	8 (16.0%)	22 (21.6%)	10 (20.8%)
Antiplatelet therapy	12 (24.0%)	42 (84.0%) ^I^	24 (23.5%)	24 (50.0%) ^D,F^
Chemotherapy				
- None	NA	NA	38 (37.3%)	23 (47.9%)
- Neoadjuvant	NA	NA	6 (5.9%)	1 (2.1%)
- Adjuvant	NA	NA	55 (53.9%)	21 (43.8%)
- Both	NA	NA	3 (2.9%)	3 (6.3%)

^1^ Five diabetes duration could not be retrieved. ^2^ One measurement is missing. ^3^ Four measurements are missing. ^A^ Significant difference from cohort 2, 3 and 4 (*p* < 0.01); ^B^ Significant difference from cohort 2 (*p* < 0.01); ^C^ Significant difference from cohort 1 (*p* < 0.05); ^D^ Significant difference from cohort 1 and 2 (*p* < 0.01); ^E^ Significant difference from cohort 1 and 2 (*p* < 0.001); ^F^ Significant difference from cohort 3 (*p* < 0.05); ^G^ Significant difference from cohort 1 (*p* < 0.001); ^H^ Significant difference from cohort 1 and 2 (*p* < 0.05); ^I^ Significant difference from cohort 1, 3 and 4 (*p* < 0.001).

**Table 2 cancers-12-00556-t002:** Complete blood count measurement data and age of the 4 cohorts at the pre-tumor laboratory measurement. CRC: colorectal cancer, T2DM: type 2 diabetes mellitus. *Mean ± standard deviation.*

Variables	Cohort 1 (Control) (*N* = 50)	Cohort 2 (T2DM) (*N* = 50)	Cohort 3 (CRC) (*N* = 102)	Cohort 4 (CRC + T2DM) (*N* = 48)
Age [years]	50.86 ± 12.20 ^A^	60.06 ± 9.35	61.45 ± 9.73	62.94 ± 9.51
Time between blood collections [months]	119.79 ± 29.17	89.53 ± 32.86	81.59 ± 32.86	88.11 ± 71.28
White blood cell count [10^9^/L]	7.30 ± 2.35	7.97 ± 2.21	7.73 ± 2.69	8.12 ± 2.70
Red blood cell count [10^12^/L]	4.80 ± 0.41	4.77 ± 0.42	4.69 ± 0.53	4.69 ± 0.46
Platelet count [10^9^/L]	270.82 ± 61.45	255.76 ± 57.75	251.06 ± 58.84	250.25 ± 58.89
Hemoglobin [g/L]	142.56 ± 12.29	140.60 ± 11.00	140.93 ± 15.89	140.81 ± 16.45
Hematocrit [L/L]	0.42 ± 0.03	0.41 ± 0.03	0.42 ± 0.05	0.42 ± 0.04
Mean corpuscular volume [fL]	87.17 ± 4.30	86.51 ± 3.93	88.80 ± 6.36	89.54 ± 4.64
Mean corpuscular hemoglobin [pg]	29.77 ± 1.75	29.55 ± 1.56	30.13 ± 2.07	30.01 ± 2.23
Mean corpuscular hemoglobin concentration [g/L]	341.60 ± 12.49	341.58 ± 12.45	338.65 ± 13.84	335.19 ± 21.39
Red blood cell distribution width [%]	13.29 ± 0.81 ^1^	13.47 ± 0.70	13.33 ± 1.02 ^2^	13.67 ± 1.01 ^3^

^1^ One measurement is missing. ^2^ 23 measurements are missing. ^3^ Seven measurements are missing. ^A^ Significant difference from cohort 2, 3 and 4 (*p* < 0.01).

**Table 3 cancers-12-00556-t003:** Annual predicted changes of platelets, platelet/hemoglobin-, platelet/hematocrit-, platelet/red blood cell (RBC) count- and platelet/mean corpuscular volume (MCV) ratios, calculated by random intercept linear mixed effect models. Reported as mean change, 95% confidence interval and *p*-value.

Variables	Control Subjects (*N* = 100)	Colorectal Cancer Patients (*N* = 150)
Platelet	−0.41% (−0.52%)–(−0.33%) *p* = 0.0011	2.18% 1.80–2.59% *p* < 0.0001
$\frac{\mathrm{Platelet}}{\mathrm{Hemoglobin}}$	−0.10% (−0.52%)–(−0.32%) *p* = 0.0033	3.77% 2.99–4.58% *p* < 0.0001
$\frac{\mathrm{Platelet}}{\mathrm{Hematocrit}}$	−0.67% (−0.85%)–(−0.53%) *p* = 0.0033	3.42% 2.72–4.16% *p* < 0.0001
$\frac{\mathrm{Platelet}}{\mathrm{RBC}\;\mathrm{count}}$	−0.40% (−0.51%)–(−0.32%) *p* = 0.0046	2.89% 2.26–3.58% *p* < 0.0001
$\frac{\mathrm{Platelet}}{\mathrm{MCV}}$	−0.68% (−0.86%)–(−0.54%) *p* < 0.0001	2.70% 2.23–3.17% *p* < 0.0001

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
