# Peer review of "Personalized Indicator Thrombocytosis Shows Connection to Staging and Indicates Shorter Survival in Colorectal Cancer Patients with or without Type 2 Diabetes"

_cancers, 2020, doi:10.3390/cancers12030556_

Round 1

Reviewer 1 Report

Many studies have reported that a raised platelet count (or thrombocytosis) may be identified as a marker of cancer, present before diagnosis. Thrombocytosis has been reported to be associated with poor prognosis in a variety of solid tumours. Moreover, association between thrombocytosis and CRC was also investigated by several studies. Different platelet count cut-off values have been proposed for the diagnosis of thrombocytosis  associated with cancer. The most commonly used cut-off value is the upper normal limit of platelets, but several other values has been suggested and some of them were even within normal range. Authors of this manuscript performed a retrospective longitudinal observational study with the inclusion of 150 CRC patients and 100 control subjects, and propose the use of a novel measure of platelet changes at an individual level, which they named “personalized indicator thrombocytosis” (PIT) useful for the detection, treatment and management of CRC patients. PIT is the ratio of platelet counts between two time points of blood collection. They found a tendency of platelet count to be higher in CRC patients versus controls at the time of cancer diagnosis. Moreover individuals with CRC had significant higher values of PIT, in comparison to controls. Significant higher PIT values were associated to more advanced staging and both local and distant metastasis, and also to shorter survival. Moreover, by using cut-off values for PIT extrapolated by ROC analyses, they found that, a PIT-based definition of thrombocytosis resulted in approximately 3-times more patients with thrombocytosis, in comparison to conventional platelet count. Thus, PIT values might indicate the individual condition of patients. Authors conclude that the assessment of PIT may be used in therapy decision and also for early cancer detection. However it should be underline that further investigations are strongly recommended. The objective of authors are very interesting; some questions should be addressed.

Major points:

  1. Are data expressed as Mean and SD or SEM? Please indicate it, in the legends and in the statistical analysis section.
  2. Put the significance in the table 1 and not only in the results. This will be useful to the reader to have direct information from the table
  3. A Linear multiple regression analysis should be performed in order to assess which is the independent predictor of PIT value
  4. Authors should report in the table the mean age (and SEM or SD) of individuals of the 4 cohorts at the time of the first blood collection.
  5. Which are the results if authors exclude individuals taking anti-platelet therapy?
  6. If authors use the equation of mean(of control individuals)+2SD to calculate the cut-off of PIT, it would be 1.17; how would be the number of patients with thrombocytosis in relation to CRC?
  7. Figure 1, 3 and 4: add the significance in the figures
  8. Authors conclude that the assessment of PIT may be used in therapy decision and also for early cancer detection. However it should be underline that further investigations are strongly recommended.

Minor points:

  1. In the introduction, objectives and aims of the study (from line 59 to 68) should be put after line 49. Then, introduce why you have decided to divide CRC individuals in with/without diabetes.
  2. A design of the retrospective longitudinal observational study should be added
  3. Line 66-68: this sentence is not clear.
  4. Line 110: Individuals with CRC is preferable than tumour group
  5. Legend Figure 1: describe the type of graph: they are box and whiskers? 95% confidence interval?
  6. Line211/212: it is not clear this sentence: “in accordance with the literature, our study demonstrated that in control subjects plateletsshowed a decreasing tendency” of that?
  7. Line 369: it is not clear the sentence
  8. Supplemental table 1: I don’t understand why in the parenthesis there is %.
  9. Legend of figure 5 is not clear.

Reviewer 2 Report

In the previous study, authors introduced the new thrombocytosis indicator-PIT, which showed more specificity than the MPV/PC and PLR. Even the sensitivity was lower, PIT could indicate the condition of patients better in the clinical cohort that authors selected. PIT is potential helpful in the clinical diagnosis. And it is meaningful to take further research with larger size clinical samples. However, a few questions may need authors to answer.

  1. Among the patients with PIT>1.12, significantly higher number of CRC related deaths were observed on the page 7. Did the authors observe any survival difference between the groups that patients (PIT>1.12) with and without anti-platelets therapy?  What the observation indicates?
  2. In the Fig 5, the hazard ration with a multiplication of 10 was used. Is this transformation reasonable?

Round 2

Reviewer 1 Report

I think that authors have appropriately replied.